# Rat sightings in New York City are associated with neighborhood sociodemographics, housing characteristics, and proximity to open public space

Michael G. Walsh

Department of Epidemiology and Biostatistics, School of Public Health, State University of New York, Downstate, Brooklyn, NY, USA

## ABSTRACT

Rats are ubiquitous in urban environments and, as established reservoirs for infectious pathogens, present a control priority for public health agencies. New York City (NYC) harbors one of the largest rat populations in the United States, but surprising little study has been undertaken to define rat ecology across varied features of this urban landscape. More importantly, factors that may contribute to increased encounters between rats and humans have rarely been explored. Using city-wide records of rat sightings reported to the NYC Department of Health and Mental Hygiene, this investigation sought to identify sociodemographic, housing, and physical landscape characteristics that are associated with increased rat sightings across NYC census tracts. A hierarchical Bayesian conditional autoregressive Poisson model was used to assess these associations while accounting for spatial heterogeneity in the variance. Closer proximity to both subway lines and recreational public spaces was associated with a higher concentration of rat sightings, as was a greater presence of older housing, vacant housing units, and low education among the population. Moreover, these aspects of the physical and social landscape accurately predicted rat sightings across the city. These findings have identified specific features of the NYC urban environment that may help to provide direct control targets for reducing human–rat encounters.

# INTRODUCTION

Rats are a prominent feature of the human landscape and can compromise public health. *Rattus norvegicus*, commonly known as the brown or Norway rat, is ubiquitous in urban settings across the United States (US), while *Rattus rattu*s, commonly known as the black rat (or roof rat), is now limited primarily to parts of the southeastern US. *Rattus norvegicus* is a burrowing species and thus often occupies the underground environs of subway and sewer systems (*Clinton, 1969*). Given its preferred environment, *R. norvegicus* does especially well in large urban centers. These rats are reservoirs for significant human

Corresponding author
Michael G. Walsh,
thegowda@gmail.com,
michael.walsh@downstate.edu

bacterial pathogens such as *Yersinia pestis*, *Leptospira* spp., *Rickettsia typhi*, *Streptobacillus moniliformis*, and *Bartonella* spp., which are responsible for bubonic plague, leptospirosis, murine typhus, rat-bite fever, and bartonellosis, respectively (*Himsworth et al., 2013*). Viral zoonoses are less commonly derived from rat reservoirs, though rats are the primary reservoir for Seoul hantavirus and a potential reservoir for hepatitis E virus (*Himsworth et al., 2013*). Some of these pathogens have been documented among rat populations in many urban centers across the US over recent decades (*Gundi et al., 2012*; *Purcell et al., 2011*; *Easterbrook et al., 2007*). Ongoing surveillance by the Zoonotic, Influenza, and Vector-borne Disease Unit at the NYC DOHMH has shown sustained prevalence of canine leptospirosis believed to be acquired through rat-associated transmission (*NYC Department of Health and Mental Hygiene, 2008*; *NYC Department of Health and Mental Hygiene, 2011*). Perhaps an even bigger public health concern is the potential for *Salmonella enterica* transmission (*Yokoyama et al., 2007*; *Swanson et al., 2007*; *Hilton, Willis & Hickie, 2002*; *Lapuz et al., 2012*). While sparse, there is also some evidence that transmission of pathogens from rat hosts to humans in northeastern cities is not just theoretical. A cluster of leptospirosis cases in a group of Baltimore children was epidemiologically linked to community rats positive for *Leptospira interrogans* (*Vinetz et al., 1996*). In addition, a high seroprevalence of *Bartonella* spp. infection (47.5% overall) was reported among injection drug users in New York City (NYC) and was believed to be rat-acquired (*Comer et al., 2001*). The investigators suggested that many of these individuals may be in greater contact with rats due to poverty, poor housing conditions, or homelessness (*Comer et al., 2001*). Consequently, rats remain a modern public health concern among many US urban populations, particularly in those urban centers that exhibit significant sociodemographic disparity in health. New York City has one of the largest rat populations of any city in the US (*Sullivan, 2008*). While some reports have documented the spatial relationship between humans and rats in NYC, these were based on specific bite incidents, limited by small numbers of reports, and are now decades old (*Childs et al., 1998*; *Clinton, 1969*). Accordingly, neither the distribution of rats in NYC, nor their relationship with the human environment, has been adequately described in this urban setting. This investigation takes a landscape epidemiology approach to describing the distribution of rats in NYC. New York City is the most densely populated urban center in the US and experiences extreme heterogeneity in sociodemographic character across its population. Moreover, the physical landscape is also spatially diverse, comprising varying structural composition and open public space on the surface, and extensive subway and sewer tunnel systems below the surface. This investigation describes the association between reported rat sightings collected from the NYC Department of Health and Mental Hygiene and specific features of the social and physical landscape of NYC. The aim of the study is to identify those features that are associated with a higher prevalence of rat sightings to determine if specific locations or environments in the city present opportunity for the prevention of human–rat encounters.

 

## MATERIALS AND METHODS

Rat data were acquired from the NYC Department of Health and Mental Hygiene (DOHMH) through the city's Open Data Portal (*NYC Open Data, 2013a*). Between January 1, 2010 and March 24, 2014, a total of 43,542 rat sightings were reported to the Department from all five of the city's boroughs. The reports were recorded by the DOHMH as a rat sighting complaint associated with a specific city address. Rat complaints were distinguished from mouse complaints in these records. Rat species was not listed for any of these reports, though it is assumed that these reports include only *R. norvegicus*.

A planimetric basemap of all NYC recreational public spaces, comprising park property, parks, playgrounds, rinks, courts, fields, ball fields, tracks, cemeteries, and recreational areas, was also obtained from the Open Data Portal (*NYC Open Data, 2013b*). Hereafter, these recreational public spaces will be collectively referred to simply as "public spaces". A shapefile of this map was used to identify the polygon area for each of these public spaces. The areas of all public space polygons were subsequently aggregated to the census tract(s) with which they intersected.

A shapefile for the Metropolitan Transportation Authority (MTA) subway system was obtained from the City University of New York Mapping Service at the Center for Urban Research (*Romalewski, 2010*). The length of each subway line segment (in meters) was aggregated to the census tract(s) with which it intersected.

Census tract data were obtained from two sources. The shapefile for the census tract polygons and the 2010 population count data were retrieved from the US Census Bureau Tiger data source (*US Census Bureau, 2014*). The proportion of individuals with less than a high school education, high school graduated, and college graduated were obtained from the American Community Survey which is also conducted by the US Census Bureau and provides socioeconomic data at the census tract level. The total number of housing units, proportion of vacant housing units, age distribution of housing structures, housing structure size, and proportion of housing units lacking complete plumbing were also obtained from the American Community Survey. The FactFinder data extraction utility was used to access the American Community Survey data (*US Census Bureau*).

## STATISTICAL ANALYSIS

Kernel density estimation was used to estimate the density of rat sightings across NYC. An isotropic Gaussian function was used for the kernel function, where the normal reference bandwidth was used as the default bandwidth (*Venables & Ripley, 2002*). The spatial object limits for the kernel density grid were obtained directly from the spatial points object, and kernel function estimated, using the kde.points() function in the GISTools package for R (*Brunsdon & Chen, 2012*). This density was then mapped with an additional layer comprising all NYC census tracts to enable the visualization of local context. Rat sightings were also aggregated by census tract and a global Moran's index was computed to assess whether significant spatial clustering by census tract was apparent across the city. Non-spatial Poisson regression was first used to model rat sightings across NYC with variables defined as below. However, the residual deviance was highly spatially

autocorrelated (Moran I statistic standard deviate $= 34.5$, $p < 0.00001$). Due to the residual spatial correlation structure in the data, this investigation used a Bayesian hierarchical Poisson model for spatial areal unit data. Specifically, a conditional autoregressive (CAR) model was used to represent random effects, with a Besag–York–Mollie (BYM) prior (*Besag, York & Molli, 1991*). This specification of the CAR modelled the number of rat sightings per census tract as a function of human population size, the proportion of the population with less than a high school education, and several features of the physical landscape: the total number of housing units per census tract, the proportion of vacant units per tract, the proportion of housing structures built prior to 1950 per tract, the proportion of housing structures with five or more family units per tract, the proportion of housing structures lacking complete plumbing facilities per tract, the average distance in meters from each rat sighting to public space and the average distance in meters from each rat sighting to the nearest subway line. The distances between each rat sighting and the nearest public space were measured and summed for all reported rats in each census tract. This sum of the distances was then divided by the total number of reported rats per census tract to give the average distance between sighted rats and public space in each tract. This procedure was repeated for the distances between all reported rats and the nearest subway lines in each census tract to give the average distance between sighted rats and subway lines in each tract. The full CAR model was structured as follows:

$$\ln(Rats_k) = X_k \beta_k + \varphi_k + \ln(Area_k)$$

where, for each census tract $k$, $X_k$ represent covariates for population size, low education ($<$high school graduate), total housing units, proportion of vacant housing units, proportion of old housing structures (built prior to 1950), proportion of large housing structures ($\geq$5 family units), proportion of housing structures lacking complete plumbing, average rat-to-open space distance, and average rat-to-subway distance, and $\beta_k$ represent the regression coefficients. The random effects are represented by $\varphi_k$ and the offset by $\ln(Area_k)$, which is the log of the total area of each census tract, $k$, and effectively models rats per unit area. The random effects were included in the model to account for spatial autocorrelation, with the BYM conditional autogressive prior used instead of the independence prior in order to account for potential residual spatial autocorrelation. Prevalence ratios (PR) for each covariate were obtained by exponentiating the regression coefficients and used to assess the associations between rat sightings and model covariates. Model inference for the regression coefficients was based on credible intervals (analogous to 95% confidence intervals with frequentist methods) obtained from 1,000 Markov Chain Monte Carlo samples. The deviance information criterion (DIC) was used to compare models that also included median income per census tract, older (pre-1920) and younger (after 1980 and 2000) housing stock, and smaller family unit housing sizes ($<$5 family units). The model with the smallest DIC is the one presented here and described above. The predicted number of rats per census tract and the residuals produced from the CAR model were subsequently mapped. The residuals were visually inspected in the map and formally tested using the global Moran's index to determine that no residual

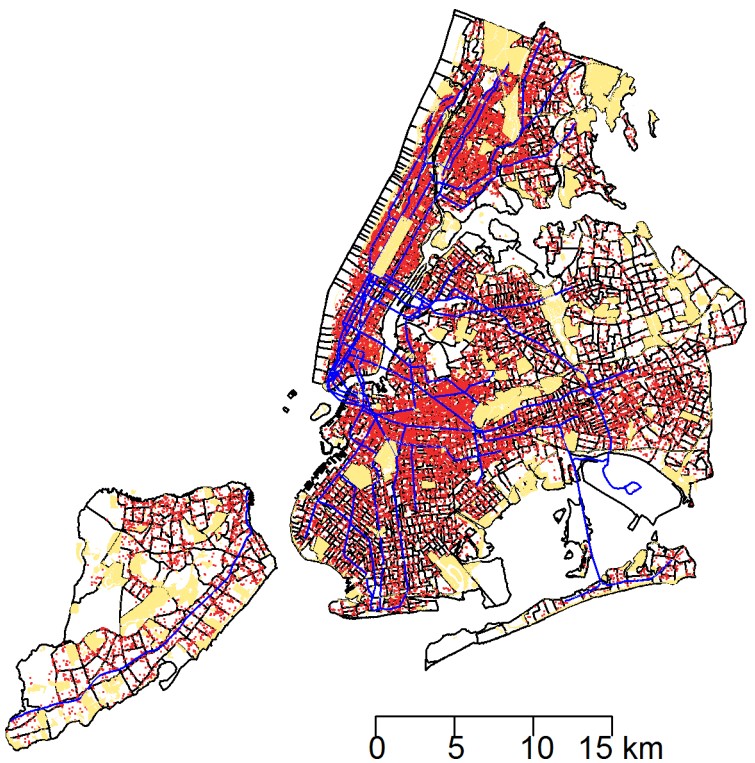

**Figure 1 The 43,542 rat sightings reported to the NYC Department of Health and Mental Hygiene are plotted with an overlay of recreational public spaces and subway lines in a map of New York City census tracts.** Rat sightings: red points. Recreational public spaces: beige polygons. Subway lines: blue lines.

autocorrelation was present. The R programming language was used for all analysis and mapping procedures (*R Development Core Team, 2010*). The CAR model was implemented with the CARBayes package (*Lee, 2013*) using the bymCAR.re() function. The spdep package was used to calculate the global Moran's index with the moran.test() function and to compute the neighborhood matrix for the CAR model using the nb2mat() function (*Bivand, 2009*).

## RESULTS

Between January 1, 2010 and March 24, 2014, a total of 43,542 rat sightings were reported to the DOHMH from all five of the city's boroughs. Figure 1 presents the map of the individual rat sightings across all census tracts along with public spaces and subway lines. Figure 2 presents the kernel density estimate of rat sightings. These maps highlight a concentration of rats in much of northern Manhattan and the South Bronx, the Lower East Side of lower Manhattan, and north-central Brooklyn. When aggregated by census tract, reported rat sightings ranged from 0 to 303 per census tract and the Moran I statistic standard deviate was 38.2 ($p < 0.00001$) indicating significant spatial clustering of rats across NYC. Figure 3 shows the concentration of rats by census tract in each of three panels juxtaposed with the average distances from rat sightings to public space and to subway lines, sociodemographics, and housing characteristics, respectively. As expected, the
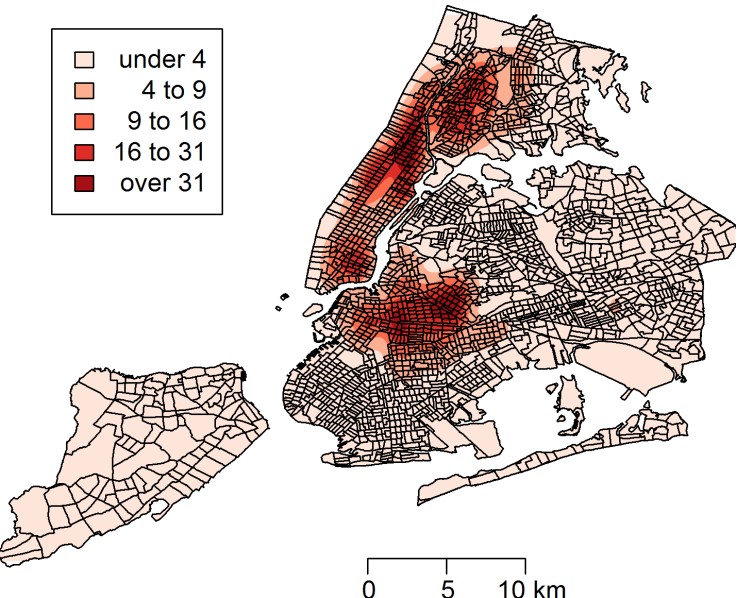

legend:
- under 4
- 4 to 9
- 9 to 16
- 16 to 31
- over 31

0    5    10 km

**Figure 2 Kernel density estimate (KDE) of rat sightings reported to the New York City Department of Health and Mental Hygiene.** The KDE is mapped with New York City census tracts.

distribution of the average distances from rats to subway lines across census tracts follows the prominent line tunnels of the MTA subway system. Census tracts with shorter average distances between sighted rats and subway lines also tend to have a higher concentration of rats. Interestingly, the distribution of the average distances from rats to public space also shares patterns with the distribution of the concentration of rats, suggesting that closer proximity to public space is also associated with a higher concentration of rats. In addition to these landscape features, high concentrations of rats appear to coincide with high concentrations of vacant housing, old housing, and low education in these maps. Estimates of the rat sighting prevalence ratios (PR) from the CAR model are presented in Table 1 along with their associated 95% credible intervals. Both the average distance to subway lines (PR = 0.996; 95% CI [0.993–0.998]) and the average distance to public space (PR = 0.94; 95% CI [0.91–0.97]) were inversely associated with rat sightings. Each 10 m increase in the average distance from subway lines resulted in a 4% decrease in rat sightings, while each 1 m increase in the average distance from public space corresponded to a 6% decrease in rat sightings. Housing factors were also associated with rat sightings, with each percentage increase in the proportion of vacant housing units corresponding to a 2% increase in rats (PR = 1.02; 95% CI [1.018–1.026]) and each 10 percentage point increase in the proportion of old housing demonstrating a 6% increase in rats (PR = 1.006; 95% CI [1.004–1.01]). The proportion of people with less than a high school education was strongly associated with rat sightings (PR = 3.97; 95% CI [1.72–5.70]), while the number of people per census tract was not. The observed rat counts and predicted rat counts for each census tract based on the CAR model are presented in Fig. 4, along with the spatial distribution of the residual deviance. Visual inspection suggests that this model is spatially homogeneous with respect to the residual deviance. Moreover, the Moran's index

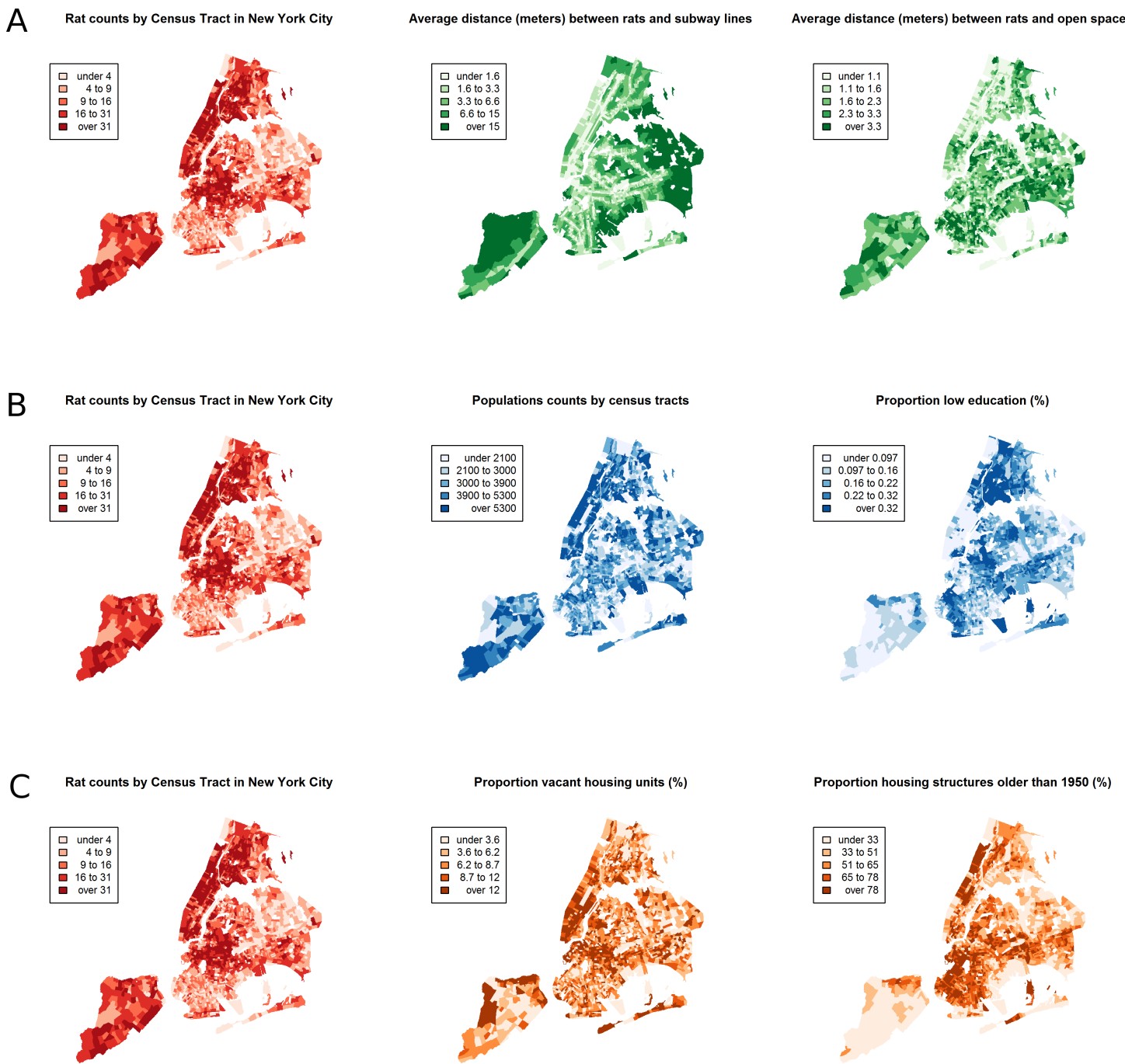

**Figure 3 The census tract distribution of rat sightings in New York City compared to the census tract distributions of landscape factors.** (A) compares rat sightings to the average distances between rats and subway lines and open public spaces. (B) compares rat sightings to the total population and the proportion of the population with less than a high school education. (C) compares rat sightings to the proportions of vacant housing and housing built before 1950. The map of rat sightings is included in each of the three panels for easier comparison to each group of mapped landscape features.

**Table 1 Adjusted prevalence ratios for the independent associations between rat sightings and the landscape factors of New York City.** The full model is a conditional autoregressive Poisson model, with Besag–York–Mollie priors. Each association between rat sightings and the independent variables are adjusted for the other variables in the model.

| Landscape characteristics | Prevalence ratio | 95% credible interval |
|---|---|---|
| Average distance from rats to subway lines (meters) | 0.996 | 0.993–0.998 |
| Average distance from rats to public spaces (meters) | 0.94 | 0.91–0.97 |
| Proportion of people with <high school education (%) | 3.97 | 1.72–5.70 |
| Total people | 1.00 | 0.99–1.00 |
| Proportion of vacant housing units (%) | 1.02 | 1.018–1.026 |
| Total housing units | 0.99 | 0.99–1.00 |
| Proportion of housing structures built before 1950 (%) | 1.006 | 1.004–1.01 |
| Proportion of housing structures with ≥5 family units (%) | 1.00 | 0.99–1.00 |
| Proportion of housing units lacking plumbing (%) | 1.78 | 0.37–30.27 |

for the residual deviance was not significant (Moran's I statistic standard deviate $= 0.78$, $p$-value $= 0.22$), thus confirming statistically the visual interpretation.

## DISCUSSION

This study identified areas of high concentrations of rats based on reported sightings in New York City. Closer proximity to subway lines and public spaces was generally found to be associated with a higher concentration of rats, as was a greater presence of older housing, vacant housing units, and low education among the population. These aspects of the physical and social landscape accurately predicted reported rat sightings across the city of New York, and may provide direct targets for reducing contact between humans and rats.

Surprisingly little research has investigated the ecology and landscape epidemiology of rat prevalence in NYC. The lack of data is even more striking considering that NYC has one of the largest urban rat populations in the western hemisphere (*Sullivan, 2008*). Nevertheless, one case-control study conducted over 20 years ago sought to identify risk factors associated with rodent bites in NYC (*Childs et al., 1998*). This study also identified proximity to subway, poverty, and population density to be associated with the risk of rodent bite in all NYC boroughs. In addition, the previous case control study identified a positive association between the number of housing units per census tract and bite risk, and a positive association between distance to parks and bite risk in Manhattan but not the other boroughs (*Childs et al., 1998*). The previous study had some limitations that don't allow direct comparison between our two studies, however. First, the case-control study counted reported rodent bites as its outcome, whereas the current investigation counts rat sightings. The former would be expected to greatly underestimate the total number of potential contacts between humans and rodents because (1) only a small minority of contacts will result in a bite, and (2) not all bites will be reported. Therefore, the previous study represents a small selection of the experience of human–rodent contact, as well as including rats and mice together in bite occurrence rather than rats alone. In addition, the small numbers of cases in the previous study likely did not have the statistical power

**Observed rat counts by census tracts**

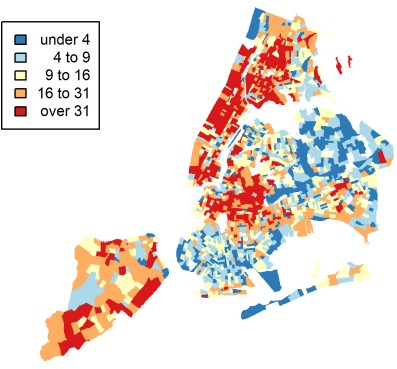

**Predicted rat counts by census tracts**

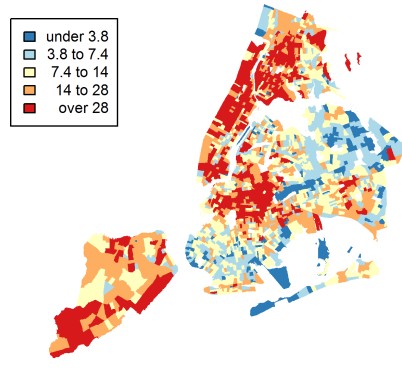

**Residual deviance by census tracts**

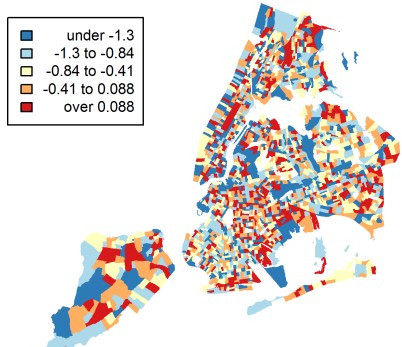

**Figure 4 Observed rat counts, based on reported sightings, and predicted rat counts from the conditional autoregressive model mapped to New York City census tracts.** The residual deviance is also mapped and shows the homogeneity of the residuals across New York City.

necessary to conduct borough-specific analyses. Indeed, none of the significant city-wide associations were also observed for every borough individually. Second, the case-control study did not account for the potential spatial heterogeneity in their logistic regression model. Nevertheless, this previous case-control study represents the most substantive exploration of the landscape epidemiology of human–rodent encounters, as measured by rodent bites, in any major US city prior to this current investigation. Interestingly, despite different measures of the human-rodent interaction outcome (rodent bites vs. rat sightings) and a different accounting of the spatial dynamics of these encounters, the current investigation still identified many of the same landscape features identified previously. However, given the much larger sample size and less restrictive outcome classification, the current findings provide more substantive support for specific features of the landscape that may be relevant for the implementation of rodent control in unique contexts.

While rats are established reservoirs for several zoonotic infections (*Himsworth et al., 2013*), very little data exist describing the pathogen population among rats in NYC. Despite the significant lack of data on pathogen prevalence in NYC rat populations and the incidence of rat-to-human transmission of those pathogens, the potential for a hidden public health burden remains given what little evidence does exist, such as the 47.5% seroprevalence of *Bartonella* spp. infection among injection drug users (*Comer et al., 2001*), or the ongoing *Leptospira* spp. transmission from rats to dogs documented by the NYC DOHMH (*NYC Department of Health and Mental Hygiene, 2008*; *NYC Department of Health and Mental Hygiene, 2011*). In the interest of reducing any such disease burden, blocking potential contact between humans and rats at specific points could be a more useful strategy than the widespread application of rodenticides, or at least a complement to such extermination programs. However, a more complete understanding of rat ecology in the city's varied urban landscape is required if we are to locate viable transmission blocks. To this end, the current report highlights some features of the landscape that may facilitate human–rat interaction, and which may present effective targets for contact control. First, proximity to public space was a strong predictor of sighted rat presence. Public spaces in NYC can accumulate garbage very quickly. Such garbage often includes discarded food, which can serve as food sources for rats, and discarded debris, which can serve as nest material or nest locations for rats (*Davis, 1953*; *Clinton, 1969*; *Sullivan, 2008*). Given the easy accessibility of most of these open public spaces, increased vigilance in their clearing and maintenance is something that could be easily adopted by current operational infrastructure (e.g., Department of Sanitation). Second, neighborhoods (or buildings) with a high proportion of vacant housing units could be targeted for more aggressive rat elimination programs. Likewise, neighborhoods of low socioeconomic status could also be similarly targeted. On the other hand, given that NYC rat populations consist predominantly or exclusively of *R. norvegicus*, and given that this is a burrowing species, proximity to subway lines is a landscape feature unlikely to be amenable to additional interventions to reduce human–rat contact. As such, the ongoing application of rodenticides will likely remain the optimal strategy for control in the subway system.

There are some important limitations attending this investigation that should be given fair consideration when interpreting the results. First, the rat counts reported here were not obtained directly through bait traps or official agency inspection of rat signs at specific locations throughout the city. Instead, all rat sightings were determined by self-report through the capture of official complaints filed with the NYC DOHMH. Therefore, there may be some measurement error in the outcome. In particular, some of the reports may be misclassified as rats, when in fact the animal sighted was a mouse. However, it is likely that such misclassification would underestimate, rather than overestimate, the true number of rat encounters for the following reason. Adult brown rats are much larger than, and quite distinct from, the common house mouse (*Mus musculus*). Confusion may arise between the two rodents when considering rat pups, which may be confused for mice. In this scenario the rat pups may be mistakenly reported as mice rather than rats. On the other hand, house mice are quite small even as adults and unlikely to be mistaken for rats. As such, where it exists, we would expect reporting error to more frequently reflect young rats mistaken to be mice, rather than mice mistaken to be rats, which would tend to underestimate the number of human–rat encounters.

Beyond the potential for misclassification of rats, the use of rat sightings as a proxy for directly measured rats may be affected by differential reporting by individuals across NYC. In other words, some individuals may choose to report a rat to the health department, while others may not. Unfortunately, we have no direct measurement to assess the characteristics of those who report rat sightings and those who don't, and how these individuals are spatially distributed across the city. Areas of high rat sighting concentration may reflect true high rat prevalence, or these areas may simply reflect a higher prevalence of people who are motivated to report a sighting. Moreover, differences in reporting distributions may be determined by sociodemographic factors such as population density or education level. Nevertheless, these characteristics were accounted for in the CAR model used in this study. Census tract area was treated as the offset, and thus the modeling of rat counts was explicitly considered as the number of rat sightings per census tract area. The total population and the percentage of low education were both included as predictors in the model. Education level was significantly associated with rat sightings, while population size was not. If rat sightings were strictly determined by the proclivity to report rather than by a function of rat prevalence, then population size would be expected to be associated with rat counts because more people overall would contribute more people who would be more likely to report sightings. On the other hand, low socioeconomic status (as measured by education level) was associated with rat counts, which would be counterintuitive if rat sighting records were, again, strictly due to reporting behavior rather than a function of rat prevalence. This is counterintuitive because these are the same areas marked by the housing characteristics that were also positively associated with rat counts. Therefore, while some influence of reporting behavior on rat sightings cannot be ruled out in the interpretation of these results, it seems likely that true rat prevalence also influences the reporting of rat sightings described in this report.

Another limitation is the census-level aggregate data. The address and geographic coordinates were recorded for each reported rat sighting, which would allow for the highest possible resolution in the spatial analysis. However, this scale was not available for any of the population-level data, thus requiring aggregation of the data at the census tract. The smaller scale of the census tract is necessarily more coarse than the larger scale observed at the level of the individual household and therefore may miss more nuanced features of rat ecology that are relevant to their encounters with humans. Nevertheless, the census tract does provide a relatively accurate description of neighborhood features that are specific to the local social and physical landscape. As such the current investigation was able to assess the relationship between rat sightings and neighborhood housing character, sociodemographics, and public space in sufficiently fine spatial scale to quantify the spatial variation across NYC.

Finally, the CAR model may have moderately underpredicted rat counts in some NYC census tracts. While there was no spatial pattern to residual deviance as noted above (Fig. 4), there was a slight preponderance of residuals with negative values. Nevertheless, the density of the residuals demonstrated a distribution reasonably centered close to zero. As such, a large bias toward underprediction is not expected, but some individual census tracts may present with a larger number of rat sightings than predicted. Future studies that measure rat presence directly will help to refine this model.

## CONCLUSIONS

In conclusion, this investigation has identified proximity to open public spaces and subway lines, the presence of vacant housing units, and low education of the population as landscape features corresponding to increased reported encounters between rats and humans. Moreover, these features present public health officials with actionable city niches that can be readily targeted for specific control initiatives, such as concerted efforts to clear public space of garbage and monitoring of vacant housing stock. In addition, results generated from this report may also be of use to investigators developing mathematical models of disease or ecologic processes. For example, these findings may be used by infectious disease epidemiologists to parameterize compartmental models used to describe the movement of zoonotic pathogens through urban populations. The findings may also be useful to ecologists developing models of urban ecology, which would most certainly benefit from a better understanding of the relationship between humans and rats. Nevertheless, it is important to note that the current study does not and cannot define specific aspects of rat ecology in NYC, which would require the collection and measurement of individual rat specimens and the detailed description of the environments from which they are collected, the behaviors exhibited, and their genomic analysis. This report is a consideration of reported rat sightings and therefore is limited as a tool of passive surveillance. Nevertheless, these findings may help to inform the development of targeted rat control programs.

### Funding

This study received no funding.

### Competing Interests

The author declares there are no competing interests.

### Author Contributions

- Michael G. Walsh analyzed the data, wrote the paper, prepared figures and/or tables, reviewed drafts of the paper.

### Supplemental Information

Supplemental information for this article can be found online at http://dx.doi.org/10.7717/peerj.533#supplemental-information.

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
