# Peer review of "Rat sightings in New York City are associated with neighborhood sociodemographics, housing characteristics, and proximity to open public space"

_PeerJ, doi:10.7717/peerj.533_

## Round 0.1 · original submission · Major Revisions

All reviewers found your study interesting and important. However, reviewers 2 and 3 raised several issues that need to be addressed before publication. The main issues, raised by both reviewers, are (i) more details about your modeling approach need to be provided, preferably in the form of the R code used to carry out the analysis, (ii) rat sightings do not necessarily equal rat prevalence, and (iii) the census boundaries in the figures make it difficult to read the figures.

·

Basic reporting

This is a well-written paper that looks at variables associated with reported rat sightings in New York City.

Experimental design

Experimental design is sufficient.

Validity of the findings

The data are analyzed and a predictive model is generated. The results draw attention to areas where rat control efforts can be more efficiently and effectively targeted.

Additional comments

Nice work. Hantavirus (line 44) is misspelled, otherwise very well written. The common name is Norway rat, not Norwegian rat, and another common name of the black rat is the roof rat.

·

Basic reporting

This manuscript describes the association between rat sightings in New York City and several potential predictors. Rats are a public health concern because they may transmit a range of bacterial and viral diseases, and they frequently share urban spaces with humans, yet their distribution and the factors that influence their population density are poorly understood. The author identifies specific factors that are associated with increased reports of human-rat interactions.

As suggested by the author, understanding these associations is important to rat control efforts, but I believe this is understating the value of the work. The author does not mention the utility of this work to epidemic modelers, who could incorporate this rat distribution model in a transmission model of rat-borne disease. More broadly, basic ecology of species that thrive alongside humans is often neglected. As urban (human) populations continue to grow, the ecology of urban (non-human) species will be of increasing importance.

I have a few suggestions that I hope will improve the manuscript:

1.) The significance of using rat sighting data instead of a direct measurement of rat population density is only partially addressed. Ideally, an observation model should be considered, if we are interested in inferring rat population density based on observations. It's easy to imagine that rats might be more reliably reported in locations where they are more rarely seen, for example. It's also possible that factors like human population density or socioeconomic factors could both affect rat population density and the probability that a rat will be seen and reported. Although I do not feel the current work needs to incorporate an observation model in order to be publishable, the possible confounding factors should be more thoroughly discussed.

2.) The subsection titled "Statistical Analysis," beginning on line 92, is almost entirely unsupported by citations. The Moran I statistic, CAR models, and BYM priors should be appropriately cited, for example. If the CAR implementation was part of this work, then details should be provided of how that was done, ideally including source code. If a package like CARBayes was used, then it should be cited accordingly (i.e., Lee 2013). If there are other details that the author can attribute to prior work, that would help readers understand the motivation behind the many decisions that go into research like this.

3.) Lines 117-121: The font used for variables here is quite different from the font used in the equation. Phi, in particular, looks like a completely different character. I encourage the author to check that a consistent font is used for publication.

4.) The results could be substantially clearer:

(a) The census boundaries, traced with black lines in the figures, dominate large portions of the map and make it difficult to judge the fill color. I would suggest not drawing any census boundaries. Having the land/water boundaries traced would be nice, but I don't know how much flexibility is provided in the R package used to draw the maps. Perhaps this could be solved with two shapefiles, one for the state of New York that would be traced, one for the census tracks that would not be. As an alternative, it may be possible to shrink the black lines by increasing the width, height and res arguments to png(), and adjusting the plot font size (cex.main, cex.axis, cex.lab) as necessary.

(b) Reproducing the “Rat counts by Census Tract in New York City” plot three times in Figure 3 is confusing. There does not appear to be a difference between the plots. Please clarify or remove the redundant plots.

(c) The lower panel of Figure 4 would be clearer if the category including 0 were white, and the over and underestimates were shown in contrasting colors. This could be done, for example, using cm.colors() in R. As an alternative to the cyan-magenta spectrum, you can specify your own, e.g., mycolorfunction = colorRampPalette(c("blue", "white", "red")).

(d) The numerical results (Table 1, and text beginning circa line 150) are somewhat confusing. An equation/explanation of how PRRs are calculated would be helpful. The terminology doesn't help; there are no “rates” and “prevalence” is unclear. I realize that there is historical baggage associated with the language. On lines 158 & 160, PRR is erroneously abbreviated RPR. In the table, it would be nice if the predictive factors could be marked, for example with an asterisk. If I understand correctly, it isn't really possible to rank them based on PRR in terms of “significance” because the magnitude of the PRR depends, in part, on the arbitrary units chosen. Is there some other way to rank the factors, e.g. explained variance?

5.) The discussion paragraph beginning on line 203 probably belongs in the introduction. It does not depend at all on the results; rather, it provides motivation and context for the work in the first place.

Experimental design

No Comments

Validity of the findings

No Comments

Additional comments

No Comments

·

Basic reporting

1) Abstract was included with PDF
2) Please add scale bars to figures 1 and 2
3) Linnean binomials do not appear to follow convention. The full binomial should be used for each new species discussed.
4) Figure 3, consider removing census track lines, it's essentially impossible to see any patterns
5) The author should consider reducing the focus on infectious diseases in the introduction and discussion.
6) There is a fair amount of information, such as the climbing habits of the two species, which is never related back to the study.

Experimental design

1) Little to no justification was given for the variables used in the model. For example, why was high school education used instead of socioeconomic data?

2) The author should focus more of the discussion on possible biases associated with using rat sightings reported to the NYC Dept. of Health. For example, consider attempting to answer the question, "what motivates someone to call the health department about rats?"

3) Please include relevant citations for R packages. If the author wrote code from the ground up, that should be indicated as well.

4) Please describe the details of the kernel density estimation.

5) In figure 4, it would be more helpful to plot the observed vs. expected frequency of reports per census block. It's very difficult to assess the model fit with the plots provided.

6) What about vertical distance to the subway tunnel? or distance from the entrance to a subway?

Validity of the findings

It seems clear that the shear size of the data set makes interpreting the results challenging. The author reported only 3 non-significant predictors, Total people, Total housing units, and Proportion housing structures with ≥ 5 family units. At the very least, please consider re-doing the analysis with people, housing units, and structures per capita or per unit area. I would also strongly recommend performing some kind of model selection.

Additional comments

The author presents an interesting analysis of an apparently under-represented topic. With some revision, this could be a nice addition to our understanding of rat populations in urban areas. However, I would strongly recommend re-focusing the modeling effort to better understand rat prevalence, as opposed to, the prevalence of rat sighting reports.

---

## Round 0.2 · Minor Revisions

The reviewers had a few more comments, as listed below.

Two additional comments from me:

1. I wanted to clarify Reviewer 3's comment about the abstract. Reviewer 3 points out that according to the PeerJ author guidelines the abstract should not be part of the manuscript document because it is inserted automatically by the submission system.

2. Thank you for providing your code to the reviewers. Please make this code available to all readers by providing it as supporting file.

·

Basic reporting

I appreciate the many improvements that the author has made to the manuscript.

To clarify my usage of the term "observation model," I was referring to an explicit, mathematical model that relates the data we have (rat sightings) to the data we would like to describe (rate prevalence). As previously stated, however, I don't feel such a model should be a requirement for publication, mostly because building a good model may be impossible with available data. I'm satisfied with the changes the author has made to the discussion regarding the limitations of working with rat sighting data.

I have two remaining concerns:

In Figure 4, almost all of the residuals appear negative. It seems possible that this is an artifact of the color palette and a skewed residual distribution, but a comparison of the observed and predicted panels also suggests that the model may consistently underpredict rat counts. Comparing the first two panels carefully is impossible, because the colors correspond to somewhat different value ranges. Also, blue means low and red means high in the upper two panels, but the color scheme is confusingly reversed in the residuals panel. Please address these points, particularly the apparent bias in the model.

Finally, and I apologize for not mentioning this previously, but throughout the paper the usage of the verb "comprise" is grammatically incorrect. Comprise does not take "of" as a preposition. E.g., Line 281 should read "Such garbage often comprises rotten food [...]"

Experimental design

No Comments

Validity of the findings

No Comments

Additional comments

No Comments

·

Basic reporting

1) Abstract still included with PDF

2) Please add scale bars for the heat map portions of Figs. 1 and 2, ie what do the various shades red correspond to.

Experimental design

All previous comments were adequately addressed.

Validity of the findings

While I still believe underlying behavioral variation in reporting may explain the significance of certain predictors, the author now provides a reasonable justification of his conclusions.

Additional comments

The manuscript was greatly improved during the revisions and, in my opinion, now represents an interesting addition to our understanding of urban rat sightings and an even more important call to arms for additional work on health burdens associated with urban rat populations. After sale bars for the heat maps in figs 1 and 2 are added, I would recommend acceptance.

---

## Round 0.3 · accepted · Accept

Thanks for making these final revisions.